# Challenges and Opportunities for Colorectal Cancer Prevention in Young Patients

**DOI:** 10.3390/cancers17122043

**Published:** 2025-06-19

**Authors:** Hyung Kim, Anna Melio, Vlad Simianu, Gautam Mankaney

**Affiliations:** Virginia Mason Franciscan Health Foundation, Seattle, WA 98111, USAvlad.simianu@commonspirit.org (V.S.); gautam.mankaney@commonspirit.org (G.M.)

**Keywords:** young-onset colorectal cancer, epidemiology, colorectal cancer screening

## Abstract

Incidence of early-onset colorectal cancer (EOCRC) is rising and comes with unique clinical, genetic, and epidemiologic features. This review explores the distinct characteristics of EOCRC, evaluates current screening modalities and guidelines, and highlights the challenges patients face across diagnosis, treatment, and survivorship. By synthesizing current evidence, we aim to raise awareness, emphasize the need for individualized screening strategies, and encourage further research into multifactorial causes driving this trend. These insights may help inform future guidelines, policy, and practice to better serve this growing patient population.

## 1. Introduction

Colorectal cancer (CRC) is the third most common cancer in both men and women worldwide and the second leading cause of cancer-related death in men and women combined, with more than 1.9 million new cases reported in the past year alone [1]. Since the introduction of CRC screening in the 1970s and 1980s, significant progress has been made in decreasing both the incidence and mortality rates of CRC in adults [2]. These advancements can be largely attributed to the widespread adoption of screening practices [3] and improved understanding of risk factors [4]. However, despite these advancements, there is a concerning upward trend in CRC incidence for young adults under the age of 50 years [5]. It is estimated that approximately 10% of newly diagnosed CRC occur in adults 50 years old or younger. As a result, the median age of CRC diagnosis shifted from 72 in the early 2000s to 66 years of age by 2016 in the United States [5,6]. This trend is also seen in many regions across the globe, including Australia, Asia, Europe, South and North America [7].

In this review, we will explore the evolving epidemiology of CRC in young patients, assess current screening modalities and guidelines, and examine the potential challenges associated with preventing early-onset colorectal cancer (EOCRC). Finally, we will explore potential solutions aimed at mitigating the rising burden of this disease in younger populations.

## 2. Early-Onset Colorectal Cancer

EOCRC is defined as colon and rectal cancers diagnosed in people less than 50 years old [8], and understanding EOCRC presents unique challenges. Current CRC screening guidelines are primarily based on age and family history of cancer, yet only a minority of EOCRC patients exhibit these traditional risk factors [5]. Furthermore, there is emerging evidence that the pathophysiology of EOCRC is distinct from CRC in older patients. For example, younger patients with CRC are more likely to present with features such as microsatellite instability, synchronous metastatic disease, tumors located distally in the rectum or left colon, as well as different profiles of mutations in the tumors [9]. These differences underscore the need to approach EOCRC not merely as a younger manifestation of the same disease but as a condition with unique pathophysiological characteristics that demand specialized attention.

Addressing the rising incidence of EOCRC requires multi-layered and proactive strategies, perhaps beyond what is currently applied to patients over 50 years of age. Some proposed measures include physicians tailoring screening protocols to better align with individual patient goals and their unique risk profiles, emphasizing the importance of updating family histories, implementing universal testing of CRC for Lynch syndrome, exploring innovative diagnostic tools such as blood testing for tumor DNA, as well as increasing awareness and education [10].

## 3. Identifying Individuals with EOCRC

There are two approaches to identifying individuals with EOCRC: screening or targeted testing. For patients with risk factors including inflammatory bowel disease, known cancer or cancer-predisposing syndromes, or familial CRC, EOCRC may be identified through earlier and more frequent colorectal surveillance with colonoscopy. Excluding patients adhering to specific screening programs, the rest of CRC diagnoses in younger patients are made after they present with symptoms and undergo diagnostic work-up.

### 3.1. Screening for Colorectal Cancer

Understanding the history of colorectal cancer screening offers important perspective, as what began as an effort to detect existing cancers has gradually shifted toward prevention and early detection through screening. In the face of the rising incidence of EOCRC, appreciating this evolution is critical—not only to recognize the limitations of current guidelines which were largely designed for older populations, but also to inform the development of screening strategies tailored to younger patients.

In the 1920s and 1930s, pioneering work by Lockhart-Mummery and Dukes laid the foundation for our understanding of CRC by demonstrating that these cancers often arise from preexisting and detectable early lesions [11]. Building on their findings, various screening modalities, including rigid sigmoidoscopies and guaiac-based fecal tests were introduced to detect CRC. However, the effectiveness of these early colorectal screening techniques had yet to be elucidated until the 1970s, when technological advances in flexible colonoscopes were incorporated into clinical practice. With the newfound ability to directly visualize and remove precancerous lesions, a new era for CRC prevention and early detection was introduced.

During the 1970s, three randomized controlled trials studying the impact of CRC screening on mortality were launched [12,13,14]. The results from all three trials demonstrated that CRC screening reduced CRC mortality. Following the publication of these trials, CRC screening was widely recognized as an effective strategy for reducing CRC mortality, and formal screening guidelines were developed across multiple professional societies (Table 1).

Over time, these guidelines have evolved to incorporate newer evidence and technologies. In response to the rising incidence of CRC in the younger population, the recommended starting age of screening for average-risk individuals was lowered from 50 to 45 years in 2021 [15]. Although this change reflects an effort to address the shifting epidemiology of the disease and to improve early detection among younger adults, it only captures a fraction of EOCRC patients as 7.1% of all new diagnoses of CRC are made in those younger than 45 [16].

There are three categories of available CRC screening tests: stool-based, blood-based, and direct visualization tests (Table 2). Each category has unique characteristics, benefits, and limitations, influencing their use in clinical practice.

**Table 1 cancers-17-02043-t001:** Screening guidelines for across several professional societies.

	National Comprehensive Cancer Network [17]	American Society of Colon and Rectum Surgeons [18]	American Cancer Society [19]	United States Preventive Services Task Force [15]	Center for Disease Control [20]	American College of Gastroenterology [21]	American College of Physicians [22]
Screening starting age for average risk	45	45	45	45	45	45	50
Screening modality for average risk							
Colonoscopy	Every 10 years	Every 10 years	Every 10 years	Every 10 years	Every 10 years	Every 10 years	Every 10 years
Fecal occult blood test (FOBT)/fecal immunochemical test (FIT)	Every year	Every year	Every year	Every year	Every year	Every year	Every 2 years
Flexible sigmoidoscopy	Every 5 years or every 10 years with FOBT every year	Every 5 years with FOBT every year	Every 5 years	Every 5 years or every 10 years with FIT every year	Every 5 years or every 10 years with FIT every year	Every 5–10 years	Every 10 years with FIT every 2 years
CT colonography	Every 5 years		Every 5 years	Every 5 years	Every 5 years	Every 5 years	
Stool DNA test	Every 3 years		Every 3 years	Every 1–3 years	Every 3 years	Every 3 years	
Other		Double-contrast barium enema every 5 years				Colon capsule every 5 years	
Increased risk factors for colorectal cancer							
Personal history of:	Adenomatous polyp, colorectal cancer (CRC), inflammatory bowel disease, cystic fibrosis, cancer requiring chemotherapy or radiation therapy to abdomen/pelvis						
Family history of:	Colorectal cancer or advanced polyp at any age	Colorectal cancer or advanced polyp before age 60 in first-degree relative	Colorectal cancer or advanced polyp before age 60 in a relative		Colorectal cancer or advanced polyp	Adenomatous polyp, colorectal cancer	Adenomatous polyp, colorectal cancer
Hereditary cancer syndrome history:	Lynch Syndrome, familial adenomatous polyposis (FAP) and other adenomatous syndromes, hamartomatous syndromes, etc.						
Screening colonoscopy interval for increased risk							
Adenomatous polyp	Every 3–10 years depending on risk and number of polyps						
Personal history of CRC	Every 5 years (following short-term surveillance colonoscopy)						
History of chemotherapy or radiation to abdomen/pelvis	Starting age 30–35 (possibly younger depending on timing of chemotherapy/radiation), every 5 years						
Inflammatory bowel disease	Starting 8 years after onset of symptoms if >1/3rd of colon is involved, every 1–3 years						
Cystic fibrosis	Starting age 30–40, every 3–5 years						
Positive family history	Starting age 40 or 10 years before earliest diagnosis, every 5–10 years						
Positive germline mutation	Generally more frequent and earlier age on onset of CRC with colonoscopy						

**Table 2 cancers-17-02043-t002:** Advantages and disadvantages of each screening method.

	Advantages	Disadvantages
Fecal immunochemical test	- Convenience to the patient - Non-invasive nature	- Needs to be completed more frequently than colonoscopy - Unable to detect preventable pre-cancerous lesions
Multitarget stool-based DNA test	- Higher specificity than fecal immunochemical tests - Non-invasive nature	- Higher false positive rate than fecal immunochemical tests - Expensive
Blood-based testing	- Specific to tumor-related DNA markers in the bloodstream - Non-invasive nature	- Not yet adopted as part of any screening guidelines - Unclear clinical utility in isolation
Colonoscopy	- Ability to examine the entire colon and rectum - Removal of precancerous lesions in a single procedure	- Requirement for a bowel preparation and possible medical sedation - Invasive nature
Flexible sigmoidoscopy	- Ability to examine distal colon and rectum - Removal of precancerous lesions in a single procedure - Reduced invasiveness, time, and cost compared to colonoscopy	- Requirement for bowel preparation (potentially less extensive than for colonoscopy) and possible medical sedation - Limited view of the colon compared to colonoscopy - Invasive nature
CT colonography	- Less invasive than colonoscopy - Low complication rate	- Requirement for bowel preparation - Exposure to radiation - Potential for incidental extracolonic findings - Associated procedural discomfort due to colonic distension with insufflation - Sensitivity and specificity rely on the adenoma size

### 3.2. Stool-Based Test

Stool-based tests are non-invasive and detect markers of CRC or precancerous characteristics in stool samples. Two commonly used stool-based tests are the fecal immunochemical test and the multitarget stool-based DNA test.

#### 3.2.1. Fecal Immunochemical Test (FIT)

The FIT detects human hemoglobin in the stool sample through antibodies specific for globin. This method of stool-based testing offers a better specificity than traditional guaiac-based testing by assessing for enzymatic activity of heme peroxidase which can be affected by diet and or bleeding from any part of the gastrointestinal tract. Although colonoscopy has a higher detection rate of precancerous lesions, a study comparing FIT to colonoscopies showed similar CRC detection rates between the two screening modalities [23]. Of note, patient participation with FIT was higher, likely due to its convenience and non-invasive nature. FIT does need to be completed more frequently than colonoscopy, usually annually compared to every ten years for normal results, and it is designed to detect colon cancer, not preventable precancerous lesions, highlighting its benefits and drawbacks. To date, there have not been any studies comparing FIT use or outcomes in those younger than 50 to the traditional, 50 years or older population.

#### 3.2.2. Multitarget Stool-Based DNA Test

Multitarget stool-based DNA test detects hemoglobin in stool as well as DNA shed by adenomas or CRC, improving its specificity versus fecal immunochemical tests [24]. However, this enhanced detection capability also results in a higher false positive rate. Another significant barrier to the adoption of multitarget stool-based DNA tests is its cost. In the United States, a multitarget stool-based DNA test costs about USD 600 per test. In a study analyzing and comparing the cost-effectiveness of the multitarget stool-based DNA test, FIT, and colonoscopy, the authors concluded that both the fecal immunochemical test and colonoscopy were more cost-effective than the multitarget stool-based DNA test [25]. These factors have limited the widespread use of multitarget stool-based DNA tests despite their advantages. As with FIT, direct comparison of use in patients under 50 years is lacking.

### 3.3. Blood-Based Testing

Blood-based testing represents a newer frontier in CRC screening, focusing on detecting tumor-related DNA markers in the bloodstream. Recently, a cell-free DNA blood-based test called Shield demonstrated greater than 80% sensitivity for CRC, along with 90% specificity for advanced neoplasia and 13% sensitivity for advanced precancerous lesions [26]. It became the first blood-based test for CRC to be approved by the United States Food and Drug Administration in July 2024. While no blood-based test has yet been adopted as part of the screening guidelines, as more data emerge supporting its efficacy and reliability, there is potential for blood-based tests to become one of the standard screening modalities.

### 3.4. Visualization Tests

Direct visualization tests involve the physical examination of the colon and rectum using advanced imaging techniques. The primary modalities include colonoscopy, flexible sigmoidoscopy, and CT colonography. The main data supporting these modalities preceded the lowering of the screening age to 45 years, and studies specifically addressing the benefits and or generalizability to younger populations are lacking.

#### 3.4.1. Colonoscopy

Colonoscopy is widely accepted as the gold standard for CRC screening, offering the ability to examine the entire colon and rectum as well as remove precancerous lesions in a single procedure. In a systemic review by Brenner et al. [27], the authors compiled data from four randomized controlled trials and ten observational studies. They concluded that colonoscopy reduced 40% to 60% of CRC incidence and death from CRC. Its comprehensive nature and diagnostic accuracy have made colonoscopy the most utilized screening tool in the United States. Its drawbacks include the requirement for a cleansing bowel preparation and medical sedation, as well as its invasive nature.

#### 3.4.2. Flexible Sigmoidoscopy

Flexible sigmoidoscopy is similar to colonoscopy, but a less invasive alternative that focuses on visualizing the distal portion of the colon. Randomized control trials demonstrate that flexible sigmoidoscopies reduce CRC incidence by 21% to 31% and death from CRC by 18% to 26% [28,29,30]. While flexible sigmoidoscopy is more limited than colonoscopy, its reduced invasiveness, time, and cost make it a viable option for some patients.

#### 3.4.3. CT Colonography

CT colonography, also called virtual colonoscopy, uses advanced imaging techniques to create a three-dimensional or four-dimensional reconstruction of the colon. The sensitivity and specificity of this technique rely on the adenoma size, with larger adenomas being more likely to be identified compared to smaller ones [31]. The advantage of CT colonography is that it is less invasive than colonoscopy with a low complication rate, but disadvantages include the need for bowel preparation, exposure to radiation, potential for incidental extracolonic findings, which may lead to otherwise unnecessary testing or overtreatment, and associated procedural discomfort as colonic distension with insufflation is still required [32].

### 3.5. Targeted Testing

While EOCRC may be diagnosed during screening or surveillance, most patients present with symptoms. EOCRC patients often present with gastrointestinal bleeding, sudden obstruction, or abdominal pain [33]. CRC diagnoses in young adults are made on average 6 months later than symptom onset due to various reasons including low level of suspicion, sense of invincibility in young adults, and lack of medical insurance [34].

### 3.6. Genetic Testing

Genetic tests may identify germline pathogenic variants or somatic mutations. Germline testing examines non-cancer cells to identify genetic changes that are present throughout the body. It identifies inherited pathogenic variants that make a person more likely to develop cancer. The cancers that they are at risk for depend on the underlying germline pathogenic variant. Somatic testing is generally performed on cancer cells and identifies acquired mutations that occurred during a person’s lifetime. All cancer arises from mutations, and individuals with germline pathogenic variants start off with a mutation, which is what increases their risk of cancer.

Current guideline recommendations include universal testing of newly diagnosed CRC for microsatellite instability and mismatch repair to identify hereditary nonpolyposis CRC [35], one of the most common hereditary causes of CRC. Germline genetic screening has recently been recommended for all CRC, not just those over 50 years [36].

The distinction between EOCRC and CRC diagnosed later in life had raised an important question of whether genetic testing should be included in a standard screening test. There is mounting evidence suggesting EOCRC may represent a distinct clinical entity with unique features. For instance, in a retrospective review of patients with EOCRC, Willauer et al. [9] found that EOCRC often presented at a more advanced stage, was more likely to occur in the left colon or the rectum, and had a higher propensity for metastatic disease. These differences may be driven by variations in tumor biology, including somatic or germline genetic mutations.

Hereditary cancer syndromes such as Lynch syndrome and familial adenomatous polyposis significantly increase an individual’s lifetime risk of colon cancer. While approximately 15% of all CRC cases are attributed to an underlying pathogenic variant, this number increases to up to 30% of EOCRC cases [37]. Furthermore, the prevalence of high-penetrance mutation associated with CRC is relatively higher among patients diagnosed with CRC before age 50 at 16% to 25%, in comparison to all ages (10% to 15%) [9].

Pearlman et al. [38] found in a prospective cohort study that one-third of EOCRC patients with identified gene mutations did not meet established genetic testing criteria for genetic testing. This suggests that many of the current guidelines which recommend screening based on family and personal history do not capture a portion of individuals at risk, particularly in the younger population. In response, the National Comprehensive Cancer Network now recommends the consideration of germline genetic testing in all EOCRC cases, reflecting the need for broader and more inclusive approaches to genetic screening.

As the molecular phenotypes associated with EOCRC are better characterized, somatic blood-based genetic testing might potentially detect high risk individuals and become a part of the screening armamentarium. New screening approaches such as circulating tumor DNA assays are being developed [39]. These tests can detect tumor-related DNA in the bloodstream, offering a minimally invasive way to monitor cancer risk. The development and application of comprehensive risk assessment tools for CRC will enhance early detection. By integrating family and personal history of cancer, known germline testing results, and other risk factors, these assessment tools will aid clinicians in identifying high risk individuals with tailored testing.

### 3.7. Trends and Impact of Screening

Among available screening modalities, colonoscopy has emerged as the most widely used screening method in the United States. It is favored because it is diagnostic and therapeutic, targets both cancer and precancerous lesions, and is necessary to confirm positive findings from other screening tests. It is the gold standard in CRC screening [17].

Over the past three decades, utilization of CRC screening has increased significantly [40], contributing to notable reductions in both incidence and mortality of CRC. However, this is driven by older individuals who have the highest rates of CRC, masking trends in the younger age group [41]. While the overall incidence and mortality for individuals older than 50 have decreased, the incidence and mortality of EOCRC have increased by 2% and 1% annually, respectively.

## 4. Prevention of Colorectal Cancer

While family history and genetic mutations are well-established, non-modifiable risk factors for CRC, there are modifiable risk factors such as sedentary lifestyle, excess body weight, and central deposition of adiposity [42]. In the context of EOCRC, special emphasis should be placed on exposures during childhood, adolescence, and young adulthood, as early life influences may significantly affect future CRC risk.

Several studies have shown that being overweight (body mass index greater than 27.5 kg/m^2^ or greater than 85th percentile) during late adolescence was associated with a 2- to 2.4-fold increased risk of developing CRC later in life [43,44]. Although a sedentary lifestyle and higher red or processed meat are recognized risk factors for average-onset CRC, evidence linking activity or diet during childhood and CRC remains limited [45], highlighting the need for further investigations. These early exposures may uncover actionable strategies for prevention. While direct causal relationship with EOCRC has not been firmly established, maintaining a healthy weight during adolescence appears to be a modifiable factor that could reduce the risk of EOCRC.

Despite the availability of various screening tools and substantial evidence demonstrating that CRC screening significantly reduces mortality, only 77.2% of adults 50 to 75 years old were up to date with CRC screening as of 2021 [46]. This leaves a considerable proportion of the population without adequate screening, delaying diagnosis and treatment. Key barriers to screening include being uninsured, lacking access to care, shorter duration of residence in the United States, lower levels of education and income, and living in less urbanized areas. Among those who were screened, patients underwent colonoscopy most commonly, accounting for an estimated 63.1% of screenings.

Efforts to increase CRC screening have focused on addressing financial, logistical, and awareness-related obstacles. One recent and significant policy change aimed at reducing financial barriers came in 2023 when the Centers for Medicaid and Medicare Services eliminated all out-of-pocket expenses for colonoscopies [47]. While an important step, it is equally important to continue to identify and address other barriers to CRC screening to ensure equitable access to screening.

## 5. Evolving Screening Guidelines

Until 2016, the recommended starting age for average-risk CRC screening was 50. However, in response to the rising incidence of EOCRC, the U.S. Preventative Services Task Force lowered the age of recommended screening to 45. The American Cancer Society also adopted this change in 2018. While lowering the screening age for CRC aims to improve early detection, it remains controversial. Many patients with EOCRC are diagnosed before age 45, meaning that lowering the screening age will not necessarily be able to capture these patients. Moreover, this younger cohort of patients will add over 20 million new patients to those in need of colorectal screening, creating a large burden on healthcare as well as potentially diverting resources away from older-age populations who still have a higher incidence of CRC, albeit decreasing over time, in comparison to those with EOCRC [48]. Cost-utility of colorectal cancer screening at age 40 shows USD 3284–11,532 per quality-adjusted life year [49]. Furthermore, cost-effectiveness could improve further when focusing on high-risk populations, as the prevalence of EOCRC is higher in these groups, leading to more significant health benefits per screening dollar spent.

To address these challenges, individualized screening strategies have been recommended: patients with first-degree relatives diagnosed with CRC at an earlier age or those with multiple relatives diagnosed with CRC are advised to undergo screening colonoscopy at age 40 or 10 years younger than the family member with earliest diagnosis, whichever option is earlier [50]. This tailored approach allows for more efficient allocation of resources and improves detection among high-risk populations which may include about 17–35% of EOCRC patients given their family history and genetic mutations [5].

As more data emerge and technology advances, promising developments are shaping the future of screening. These include risk prediction models such as scoring systems to estimate EOCRC risk [51], as well as possibility of integration of artificial intelligence to stratify risks and inform screening recommendations [52], particularly for patients who fall outside traditional guidelines. With continued research, truly individualized screening based on personalized risk profiles may become standard practice.

Among various proposed strategies to reduce EOCRC, two key factors are associated with higher rates of CRC screening [53]: regular access to preventative care and directly receiving a physician’s recommendation for screening. In that regard, primary care providers play a vital role. Although there is no one specific strategy that has been shown to enhance screening, the use of multiple strategies including electronic medical record use, reminders, audit and feedback reports, and dedicated staff that are associated with improving colorectal screening uptake in patients [54]. By discussing patients’ risk factors, emphasizing the importance of early detection, tailoring recommendations to individual patients’ unique preferences and needs, as well as utilizing multiple avenues for patient education, primary care providers can aid in early detection of EOCRC.

Furthermore, raising awareness among both patients as well as providers is crucial. Technological advancements and the growing influence of digital platforms offer new opportunities to engage with patients. For instance, electronic media has demonstrated higher patient engagement and also led to an increased likelihood of completing screening [55]. Social media, in particular, has become an effective tool for spreading awareness about CRC, with the increasing attention of CRC on social media [56]. Developing targeted campaigns and educational materials for social and electronic media may further boost colorectal screening rates, especially among younger populations.

## 6. Treatment and Survivorship

In addition to the challenges detailed above regarding screening, prevention, and treatment of EOCRC, there exist significant challenges regarding the management of CRC survivorship. Many of these challenges are under-researched in the realm of young patients, despite arguably having a larger impact.

### 6.1. Sexual Health

The treatment of CRC has significant impacts on sexual function. The management of CRC for individual patients often includes a combination of chemotherapy, radiation therapy, and surgery. These factors individually, and in concert, affect a patient’s sexual functioning, including impacting on sexual desire, intercourse frequency, ability to experience orgasm, and overall sexual satisfaction.

The impact of sexual functioning on patients with CRC was evaluated in a review of 487 patients, two years following treatment of their colon or rectal cancer. Over half of patients reported decreased sexual desire and intercourse frequency, while nearly half reported a decrease in the ability to achieve orgasm. These impacts were more significant in patients with rectal cancer compared to patients with colon cancer, and in those with reported fecal incontinence. Only 20% of men and 11% of women discussed having conversations about sexuality and sexual health with a member of their cancer care team [57].

The impact of sexual functioning is also at times gender-specific; female patients report decreased libido, arousal, lubrication, and report dyspareunia, whereas male patients cite impotence, erectile dysfunction, and difficulty with ejaculation [58]. In a qualitative study evaluating male patients two years following treatment found that patients, authors reported a lack of pre-operative education surrounding expectations of the sexual impact of treatment and surgery [59].

Certain populations’ sexual functioning is disproportionately impacted by CRC treatment. There is a paucity of data on the sexual functioning and satisfaction in patients for whom anal receptive intercourse is their primary method of sexual activity. In a study of the experiences of gay and bisexual men following prostate cancer treatment, these men reported increased psychological distress, decreased sexual intimacy, decreased self-esteem, and significantly lower sexual functioning and quality of life, in comparison to age-matched controls [60]. Additionally, the comparison of the experiences of this population receiving prostate cancer treatment cannot be directly applied to patients who undergo CRC treatment.

The management of sexual dysfunction in patients following treatment is multifactorial. As discussed previously, a significant cause for distress in patients is the lack of pre-treatment education surrounding possible impacts on sexual functioning [57]. Education for patients either pre-operatively, or in the early recovery periods of treatment, would provide a helpful framework to continue to promote sexual health [61]. It is also imperative to continue to discuss sexual functioning as a measure of recovery and quality of life in order to address problems as they occur.

A review of general post-cancer medical care found that the large majority of general post-cancer care is managed by primary care providers, and that education provided to providers in regard to screening for and managing sexual health dysfunction could improve patient support [62]. The specific management of sexual dysfunction varies based on the individual issue; however, multiple therapies exist. Referral to a sexual health provider for evaluation is often the first step. Management includes, but is not limited to: psychosocial interventions, topical or oral medications to assist with sexual functioning, hormone replacement, pelvic physical therapy, and even surgical options such as implants for erectile dysfunction [61].

### 6.2. Fertility

In addition to concerns about sexual functioning, younger patients who are diagnosed with CRC also have additional considerations of fertility preservation in comparison to the older population.

The tenants of management of CRC include chemotherapy, radiation therapy, and surgical resection, all of which have a variable impact on fertility. Additionally, as detailed above, there exists a risk of decreased sexual functioning or erectile dysfunction following CRC treatment interventions, which also impacts fertility [61].

Chemotherapeutic regimens for CRC are typically 5-Flurouracil (5-FU)-based. 5-FU itself has almost no effect on reproductive function or fertility [63]. Oxaliplatin, often used in combination with 5-FU as a chemotherapeutic regimen, has moderate gonadotoxic effects [64]. A retrospective review demonstrated that 16% of patients who had FOLFOX chemotherapy were persistently amenorrheic one year after treatment [65].

Pelvic radiation therapy leading to premature ovarian failure has been well described in the literature. Premature ovarian failure refers to undetectable anti-Mullerian hormone levels [64]. Over 90% of patients who undergo radiation therapy for rectal cancer experience premature ovarian failure [66,67].

The impact of radiation therapy as compared to chemotherapy is evident in a study comparing the incidence of persistent amenorrhea in patients younger than 40 who underwent treatment for either colon or rectal cancer. The incidence of amenorrhea was 4% in those with colon cancer, who had chemotherapy, vs. 96% of those with rectal cancer who completed chemoradiotherapy [68].

The impact of surgical treatment on fertility is not as easily quantifiable. Data on the impact of fertility by surgical resection is more studied in cases of benign disease, such as ulcerative colitis, over CRC. A study of nearly 3000 women and matched controls in Sweden following colectomy for inflammatory bowel disease demonstrated decreased fertility following surgical resection, with lower rates of infertility when the rectum was left intact [69]. Another study demonstrated a nearly 3-fold increase in rates of infertility following ileal-pouch anal anastomosis for ulcerative colitis [70]. Pelvic surgery increases the risk of pelvic adhesion formation which has been associated with decreased fertility [71]. Importantly, the success rates of in vitro fertilization following surgical resection for ulcerative colitis were not diminished [72].

Studies report a low rate of fertility counseling among female patients prior to CRC treatment, with between 15 and 25% reporting having received appropriate counseling [73]. When compared with patients who did not receive counseling, young female patients with new cancer diagnoses who underwent fertility preservation counseling noted decreased long-term regret surrounding fertility, and improved quality of life scores. This was true whether or not the participants elected to pursue methods of fertility preservation [74].

Discussions and concerns about fertility are not limited to female patients, as the risk of sexual dysfunction following CRC treatment in male populations is significant. Options for fertility preservation are outside the scope of this review, but are based on oocyte or embryonic cryopreservation.

Further investigation into not only the impact of fertility on the CRC population, but also its mitigation, are needed. As younger populations continue to be diagnosed with CRC at higher rates, it is imperative that discussions of fertility preservation are included in patient care and management.

### 6.3. Post-Treatment Surveillance

Following treatment for CRC, patients require ongoing post-treatment surveillance, as they are at a higher risk for developing another primary CRC [75]. Specific surveillance methods and frequency depend on the stage of the cancer. Standard approaches consist of history and physical exam, colonoscopy, and, for later-stage cancers, carcinoembryonic antigen (CEA) and chest, abdomen, pelvis CT scans [26]. After the acute peri-operative period, current guidelines recommend surveillance colonoscopy every 5 years for patients with personal history of CRC, which is more frequent than every 10 years for average-risk patients.

Despite presenting with advanced-stage disease compared to patients with average-onset CRC, individuals with EOCRC tend to have longer disease-specific survival [76]. Furthermore, patients with EOCRC did not have an increased risk of cancer compared to patients with average-onset CRC in short-term follow-up [77]. Consequently, existing guidelines for post-treatment surveillance might be appropriate for patients with EOCRC as well.

However, frequent surveillance colonoscopies can impose significant financial, personal, and healthcare resource burdens, particularly for EOCRC patients who face longer disease-specific survival and may therefore undergo more surveillance over their lifetime. Garg et al. estimated the increased direct medical cost for EOCRC versus average-onset CRC, reporting average annualized cost of USD 50,216 versus USD 37,842 during the initial phase following diagnosis, and USD 8361 and USD 5014 for continuing surveillance, respectively [78]. As the population of EOCRC survivors grows, there is a need to balance the benefits of early detection with potential burdens of ongoing surveillance to ensure equitable and effective care for patients with CRC.

## 7. Conclusions

CRC remains a significant public health challenge, especially with evolving trends highlighting the growing burden of EOCRC. While advancements in screening and treatment over time have led to a decline in incidence and mortality among older adults, the increasing incidence of EOCRC demands a more nuanced approach. This trend underscores the need for individualized screening strategies, heightened awareness, ongoing research into genetic, environmental, and lifestyle factors contributing to this phenomenon, as well as increased awareness to address the unique challenges to EOCRC adequately.

The latest guideline update lowering the screening age to 45 reflects efforts to address this growing concern, but many young patients with EOCRC remain undiagnosed due to limitations of traditional screening criteria. A multifaceted approach combining prevention, early detection, personalized care, and symptom identification is vital to reducing the burden of CRC across all age groups.

## Data Availability

No new data were generated or analyzed in this study.

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
