# Peer review of "Challenges and Opportunities for Colorectal Cancer Prevention in Young Patients"

_cancers, 2025, doi:10.3390/cancers17122043_

Round 1
Reviewer 1 Report (New Reviewer)
Comments and Suggestions for Authors
I think the article is well written and all the reviewers' requirements are adequately met in the revised version.
Author Response
Comment 1: [I think the article is well written and all the reviewers' requirements are adequately met in the revised version.]
Response 1: [Thank you for reviewing our article.]
Reviewer 2 Report (New Reviewer)
Comments and Suggestions for Authors
Accept in current form.
Excellent work
Author Response
Comment 1: [Accept in current form. Excellent work]
Response 1: [Thank you for reviewing our article.]
This manuscript is a resubmission of an earlier submission. The following is a list of the peer review reports and author responses from that submission.
Round 1
Reviewer 1 Report
Comments and Suggestions for Authors
This paper presents a breadth of information about CRC - risk, screening, treatment, side-effects of treatment - much of which is known (and has been summarized in the literature) and not specific to EOCRC. It is not clear that this review provides any new information or novel interpretation of existing literature on EOCRC. I suggest the authors pair this down with a focus on EOCRC and offer new perspective or paths forward for new research.
Reviewer 2 Report
Comments and Suggestions for Authors
General
The review of Kim et al. provided a comprehensive picture about the prevention of early onset colorectal cancer. The authors led a clear line of thought from the epidemiology, through screenings to the survivorship of colorectal cancer. In general, the paper fulfilled its objectives, however, some minor concern remained.
Minor remarks
- The paper cited only US epidemiological data. Beside this global data should be also referred (e.g. GLOBOCAN).
- Based on the adjustment of the available literature, the authors concluded, that screening tolls may be performed only in case of positive anamnestic data (previous family occurrence, specific mutations). Is there any data about the amount of the potentially affected populations? Cost-benefit analysis should also be performed.
Summary
The recent comprehensive report summarized our current knowledge properly. It will be worthful for publication after minor revision.
Reviewer 3 Report
Comments and Suggestions for Authors
This manuscript reports the rising incidence and the clinical features of early-onset colorectal cancer, and highlights the need for personalized screening strategies and improved survivorship care to address the challenges faced by younger patients. I have the following comments.
1.It is recommended to use tables or charts to summarize the advantages and disadvantages of each screening method.
2.The title emphasizes prevention, but the section 4 of the main text lacks sufficient elaboration. For example, actionable strategies could be proposed in response to the current challenges.
3.Line 95: The term was referred to as “multitarget stool-based DNA test” earlier, but later appeared as “multitarget stool DNA test.” It is recommended to use a consistent term throughout the manuscript.
4.Line 250:The term “competing”in the text appears to be a spelling mistake and should be corrected to “completing”.